# Entanglement-inspired frequency-agile rangefinding

**Weijie Nie** ⓘ ✉, **Peide Zhang, Alex McMillan, Alex S. Clark** ⓘ **& John G. Rarity**

Entanglement, a key feature of quantum mechanics, is recognized for its non-classical correlations which have been shown to provide significant noise resistance in single-photon rangefinding and communications. Drawing inspiration from the advantage given by energy-time entanglement, we developed an energy-time correlated source based on a classical laser that preserves the substantial noise reduction typical of quantum illumination while surpassing the quantum brightness limitation by over six orders of magnitude, making it highly suitable for practical remote sensing applications. A frequency-agile pseudo-random source is realized through fiber chromatic dispersion and pulse carving using an electro-optic intensity modulator. Operating at a faint transmission power of 48 $\mu$W, the distance between two buildings 154.8182 m apart can be measured with a precision better than 0.1 mm, under varying solar background levels and weather conditions with an integration time of only 100 ms. These trials verified the predicted noise reduction of this system, demonstrating advantages over quantum illumination-based rangefinding and highlighting its potential for practical remote sensing applications.

Quantum entanglement has been widely explored for its potential to improve performance in the fields of communication[1,2], computing[3,4], sensing[5–9] and imaging[10,11]. In the field of quantum metrology, the method offers advantages including enhanced precision[12–14] and noise suppression[15–19], which are particularly critical in remote sensing under challenging conditions, such as strong background and increased transmission loss. Since Lloyd's pioneering work in 2008[20], quantum illumination has garnered significant attention for its superior noise reduction capabilities in noisy and lossy environments. This improvement is primarily attributed to the entanglement and correlation properties of photon pairs[21,22]. Lloyd's theory projected that using $n$-dimensional entangled photon states could reduce noise compared to unentangled single-photon states. Subsequent studies[18,23,24] have explored splitting quantum sources into $n$ frequency modes, allowing quantum systems to filter out unentangled noise via energy-time entanglement measured using joint detection, and demonstrating an enhanced signal-to-noise ratio (SNR).

However, applying quantum illumination to practical target detection presents significant challenges[25,26]. Generating, maintaining and detecting entangled photon pairs are more complex processes than their classical analogues, making quantum setups less robust. For instance, entanglement-based detection can require quantum memory to store the reference photon, complicating the detection platform and requiring prior knowledge of the target's distance, which limits its applicability in rangefinding. The target in any rangefinding demonstration is a classical object that is either there or not, which in the language of quantum information means it is in the computational basis. As such, correlation measurements (without entanglement) can retain the key noise-reduction benefits while simplifying the system[24,27,28]. These advancements eliminate the need for pre-existing range information and quantum memory, making the setup more practical for distance measurement. Typical setups generate correlated photon pairs using a nonlinear crystal[18,24,29,30] or semiconductor optical chip[15], sending one photon of each pair (signal) to the target while keeping the other (idler) as a reference. Range can then be

Quantum Engineering Technology Labs, H. H. Wills Physics Laboratory and School of Electrical, Electronic and Mechanical Engineering, University of Bristol, BS8 1FD Bristol, UK. ✉e-mail: weijie.nie@bristol.ac.uk

measured using time differences between the two photons of each pair. Uncorrelated noise is filtered out via energy-time correlations, noticeably boosting detection sensitivity.

Despite these improvements, the brightness of photon-pair sources remains a significant limitation. Brightness is critical for remote target detection, as the number of detected back-scattered photons diminishes quadratically with distance, and at longer ranges, this can extend to a quartic relationship due to light divergence. The brightness of commonly used spontaneous parametric down-conversion (SPDC) sources is inherently limited by multi-photon emission, which when avoided results in low emission rates. Semi-conductor quantum dot sources offer more efficient photon genera-tion, but extracting photons is still challenging due to the high refractive index of surrounding materials[31]. Additionally, the maximum detection rate is constrained by the number of photons per time resolution window (typically > 100 ps). For example, with correlated photon pair emission into $n$ modes and detection using $n$ single-photon detectors, the maximum count rate can reach $10^{10}$ photons per second (assuming $n = 10$ and 0.1 pairs per detection window). This limits the maximum system loss to ~ 80 dB, considering an overall system efficiency of ~ 20 dB, which restricts the practical use of quantum illumination in remote sensing. In contrast, classical sources can easily achieve many orders of magnitude higher brightness levels[32,33], making them more suitable for long-range detection. For example a 600 mW average power laser with a repetition rate of 500 kHz results in $10^{13}$ photons per pulse, has been reported for ima-ging over 200 km[33].

Recent research has explored the idea of using energy-time correlations from coherent sources to maintain quantum-like advantages[34–36]. However, these systems rely on nonlinear proper-ties, which introduce a similar complexity to quantum sources and confine the detection range to be within the coherence length. As a result, all of them are demonstrated under controlled laboratory conditions and are difficult to extend to real-world remote sensing.

To address these limitations, we have developed a robust sys-tem that generates energy-time correlations without requiring a frequency conversion process, by modulating a classical source. This quantum-inspired rangefinding employs classically correlated optical sources while preserving most of the noise-reduction bene-fits of quantum rangefinding, which uses entangled-photon sources. To achieve the benefit of entanglement requires operating at the time-bandwidth limit at low photon numbers per timing window, limiting operational distance[22]. This approach validates our theore-tical model (Eq. S10 in Supplementary Section II) where we find that the SNR simplifies to

$$\text{SNR} = \frac{\mathcal{S}}{\sqrt{\mathcal{S} + \mathcal{D}/n + \mathcal{B}/n}} . \tag{1}$$

where $\mathcal{S}$, $\mathcal{D}$ and $\mathcal{B}$ are the total correlated signal counts, dark counts and background counts across all channels. We clearly see that the dark and background counts are suppressed by using multiple channels $n$, resulting in an enhanced SNR (see Supple-mentary Section II for full derivation). Using a three-channel ($n = 3$) frequency-agile correlated source based on fiber chromatic dispersion, we demonstrate noise reduction through field trials under various weather conditions over distances of more than 150 m. With the transmitter power attenuated to less than 50 μW the system shows potential for eliminating crosstalk in multi-use conditions, or for covert illumination, while demonstrating significantly higher brightness than entangled photon sources. Notably, the source power can be easily increased, making it suitable for longer-range measurements and showing that the brightness limitations of quantum systems can be bypassed without sacrificing noise suppression.

## Results

Energy-time correlations in our quantum-inspired source are built up by stretching femtosecond laser pulses to nanosecond scale using fiber chromatic dispersion then pulse carving to create three fre-quency channels as described in the Methods section and illustrated in Fig. 1. Figure 2a and b display the characterization of the created cor-relation with respect to time and energy, respectively. The gray area with a black line represents the stretched pulse of the primary fs-laser before temporal modulation, while the colored areas correspond to the three selected channels after electro-optic intensity modulator (EOIM) modulation. Figure 2a demonstrates the time-domain selection and separation of the three channels by applying different time delay windows to the stretched pulse. Figure 2b shows the energy separation of the three channels, characterized by their spectra with central wavelengths of 1532.2 nm, 1562.5 nm, and 1583.7 nm. The direct cor-respondence between the pulse time sequence and the increasing wavelength spectra demonstrates the successful creation of energy-time correlation across the three channels. The three wavelengths are selected using a pseudorandom sequence of length 10 μs to allow unaliased rangefinding up to 1.5 km distance. After successfully creating three separate wavelengths by time-gating, the short pulse is reformed in a grating-based compressor.

The quantum-inspired correlation source can achieve an average power of 147.5 μW before the fiber splitter, corresponding to $1.15 \times 10^{15}$ photons per second. This represents more than 60 dB improvement compared to conventional quantum sources of $1 \times 10^9$ photons per-second assuming 0.1 photon pairs per time window (100 ps) on average[24]. This significant enhancement in source brightness has enabled us to extend the detection distance from several meters in the lab[24] to field trials conducted between two buildings, separated by a distance of over 150 m.

The results from night-time measurements, where photons were scattered back from the external wall of Wills Memorial building (WMB) with an integration time of one second, are shown in Fig. 3. Figure 3a presents the photon-counting histograms of return photons in three channels within one source period of 20 ns. The leftmost peaks of the three channels are the main return signals and are well aligned with each other, indicating that the energy-time correlation has been effectively concealed by the grating-based compressor. The right peaks are the breakthrough from the EOIM caused by reducing the repetition rate from 100 to 50 MHz. This leakage arises from the relatively low EOIM extinction ratio, which is due to the broadband spectra in each channel. This issue could be mitigated by increasing the number of channels, thereby narrowing the spectral width of each channel. Figure 3b shows a 200 ns excerpt of a 10 μs long photon-counting histogram, triggered by the start of the random pattern. Transmission losses scattered from the Lambertian target were cal-culated to be 98.1 dB in this experiment, resulting in a maximum count rate of only 6 photons per time bin, some of which may include background noise or detector dark counts. The smaller peaks with counts around one photon are due to leakage from the EOIM, dark counts, and background noise. By post-selecting the encoded timing information, we can identify that peak 1 corresponds to the echo sig-nal, peak 2 is EOIM leakage, and peak 3 is detector dark counts. Given the night-time conditions, the background noise level was low and can be considered negligible.

Distance information was extracted from the photon-counting histograms of the return signal using the coincidence-based correla-tion method. The coincidence counts in each channel were calculated by cross-correlating the normalized reference, made up of the random channel sequence repeating for the chosen integration time, with the echo photon counts for each channel, resulting in three-channel coincidence counts shown in Fig. 3c. From this data the round-trip time was estimated from the location of the highest peak, and the corresponding range was calculated using the formula $r = ct/2$, where

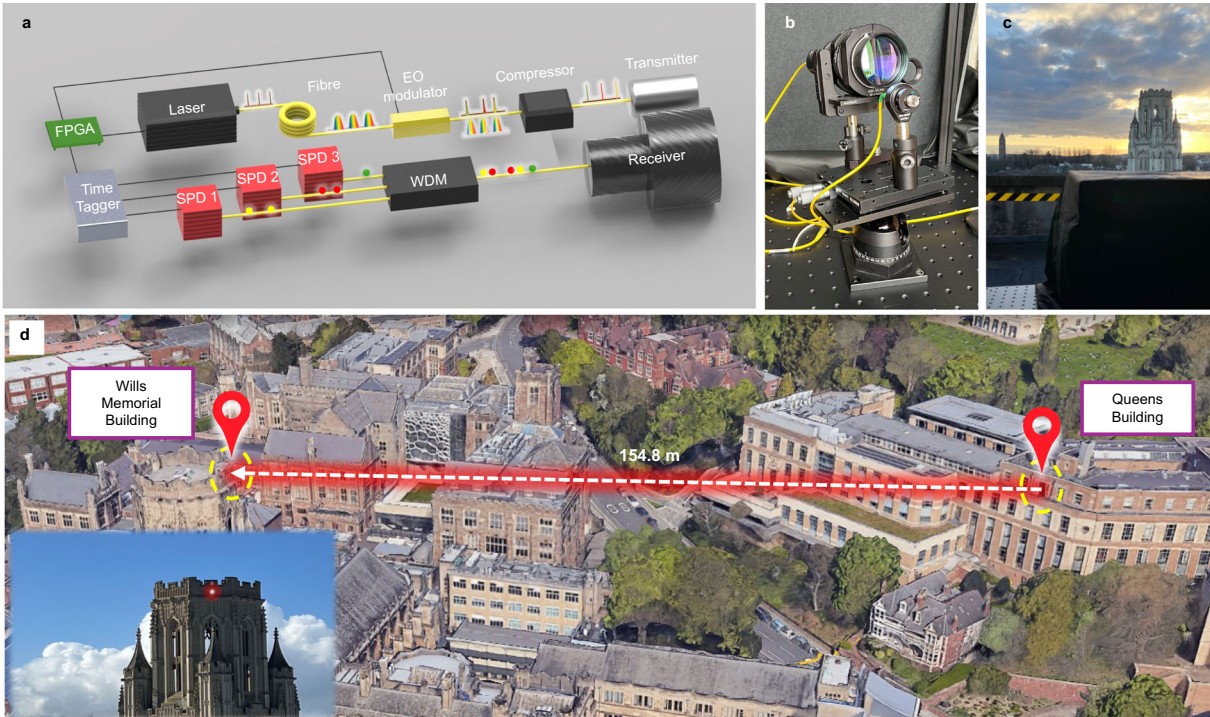

**Fig. 1 | Experimental setup. a** Experimental setup for the rangefinding field trial. EO modulator: electro-optic intensity modulator; FPGA: field-programmable gate array. **b** The front view shows the transmitter and receiver on the balcony, connected to the lab by two single-mode fibers. **c** The back view depicts the black box housing the transmitter and receiver aimed at the target (the external wall of the Wills Memorial Building (WMB)). **d** A 3D Google map shows the distance between the balcony (Queen's Building) and the target (WMB), with the bright white spot in the inset indicating the illumination region of the target. Underlying map from Google Earth, captured on 19 July 2024 (Data: SIO, NOAA, U.S. Navy, NGA, GEBCO, Airbus, Landsat/Copernicus, IBCAO; © Google).

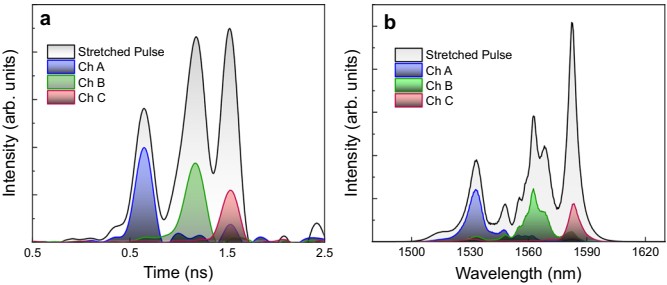

**Fig. 2 | Energy-time source characterization. a** Time-domain pulse traces showing the separation of three selected channels (colored regions) from the stretched pulse (gray area with black line) originating from the initial fs-laser pulse. **b** Spectral separation of the three energy channels (colored regions) derived from the original classical light source (gray area with black line).

$c$ is the speed of light and $t$ is the round-trip time of the illuminating photons. Magnifying the peak region in the inset, several periodic side peaks can be observed. The relatively high side peaks are due to the 50 MHz pulse repetition period, while the smaller peaks arise from leakage of the 100 MHz pulse. The measured peak (with a time bin of 50 ps) was fitted using a Gaussian function (see Fig. 3d), centered at a distance of 154.81822(5) m. This fitting improves the measurement precision to ± 48 μm, representing the statistical uncertainty in locating the peak centroid and overcoming the limitations set by detector jitter. In contrast the single shot depth resolution is determined by the full width at half maximum (FWHM) of combined the three-channel histogram shown in Fig. 3d, which is 25.7(1) mm. This value indicates how well two targets at different depths can be distinguished. Additionally, the unambiguous range was extended from 3

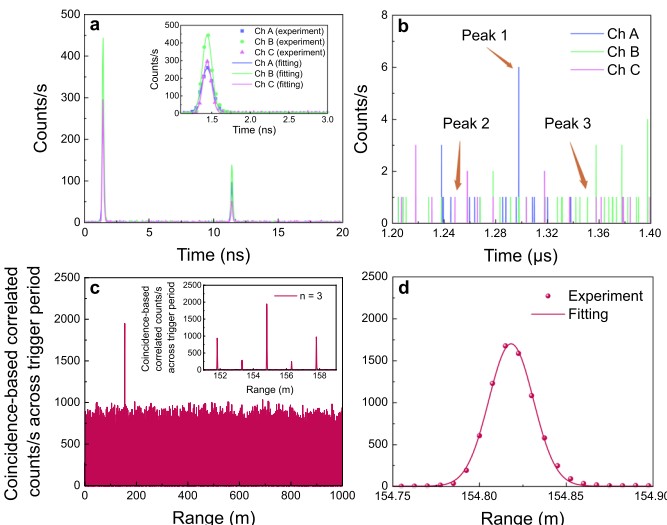

**Fig. 3 | Nighttime characterization results. a** Return photon-counting histograms for each channel (A, B, and C) triggered at the source repetition rate of 50 MHz, displayed within one source period of 20 ns with one second integration time. Inset: magnified view of the primary signal peak. **b** A 200 ns excerpt of a return photon-counting histogram triggered at the pattern repetition rate of 100 kHz, integrated over one second. Peaks 1, 2 and 3 correspond to echo signal photons, modulator leaky photons, and dark counts, respectively. **c** Coincidence-based cross-correlation between the normalized reference electrical signal and the detected echo photons back from WMB, with an integration time of one second. The highest peak at 154.81822(5) m marks the measured range. Inset: expanded view of the main peak, showing periodic pulses from the original source. **d** The main peak, further magnified from the inset of **c**, fitted with a Gaussian function (red line).

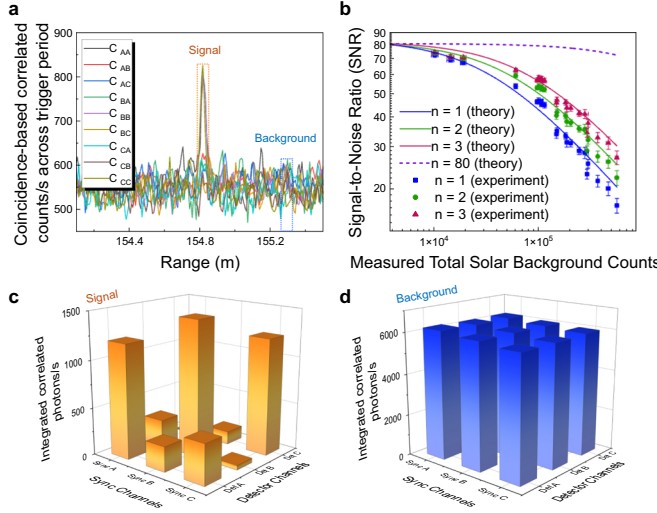

**Fig. 4 | Daytime enhancement to SNR. a** Coincidence-based cross-correlation peaks $C_{ij}$ between the normalized reference channels $i$ and detected photon channels $j$ ($i, j \in A, B, C$) collected in bright daylight (sunshine falling on WMB) and one second integration time. **b** Signal-to-noise ratio for one-channel ($n = 1$), two-channel ($n = 2$), and three-channel ($n = 3$) platforms, integrated over one second, as a function of the detected total solar background. Symbols represent the experimental data with error bars showing the standard deviation measured from 60 independent range measurements, with the fitted theoretical model shown as colored curves. The purple dashed line shows the simulated results for 80 channels using commercial dense wavelength division multiplexing (DWDM).
**c, d** Correlation tomography extracted from the orange signal region and blue noise region in **a**, respectively.

m to 1500 m, set by the length of the random channel sequence. This breaks the limit of a periodic source on maximum unambiguous detection distance[37], caused by the high repetition rate of 50 MHz, and could be further extended using a longer random sequence or even a real-random sequence.

To experimentally validate the theoretical model that shows a significant noise reduction effect from using multiple energy channels, we compared a one-channel ($n = 1$), two-channel ($n = 2$) and three-channel ($n = 3$) system under varied solar background levels using data from three single-photon detectors. Since the periodic peaks are primarily due to the source repetition rate, we focused on the range peak region within a 1.5 m (10 ns) window, excluding the influence of the periodic signal.

First, coincidence-based correlated counts $C_{ij}$ were calculated between the normalized reference channel $i$ and detected photon channel $j$ ($i, j \in A, B, C$) using data collected during the daytime with 1 s integration time, as displayed in Fig. 4a. Here, correlated elements occur when $i = j$, and uncorrelated elements occur when $i \neq j$. For single-channel measurements ($n = 1$), no energy-time correlation exists, meaning all elements are considered:

$$C_{total, n=1} = C_{AA} + C_{AB} + C_{AC} \\ + C_{BA} + C_{BB} + C_{BC} \\ + C_{CA} + C_{CB} + C_{CC}. \quad (2)$$

For two-energy channels ($n = 2$), two of the three photon detection channels (e.g., A and C) are treated as one combined channel. The correlated counts between this combined channel and the third channel (e.g., B) are calculated as follows:

$$C_{total, n=2} = (C_{AA} + C_{AC} + C_{CA} + C_{CC}) + C_{BB} \quad (3)$$

Finally, for three-energy channels ($n = 3$), the correlated counts for all three channels are given by:

$$C_{total, n=3} = C_{AA} + C_{BB} + C_{CC}. \quad (4)$$

As illustrated in the correlation tomography of Fig. 4c and d, the signal and noise counts were extracted from Fig. 4a by integrating the peak region, with the average noise level subtracted (orange area) and the fluctuating region, estimated as the average noise level multiplied by the integration time (blue area). In Fig. 4c, the signal counts primarily come from correlated elements ($i = j$), indicating low crosstalk across multiple energy channels. Uncorrelated counts ($i \neq j$), which are primarily induced by the channel crosstalk and time jitter, can be minimized by increasing the delay between adjacent channels. As channel orthogonality and independence are crucial for the developed energy-time correlations, low crosstalk and minimal timing jitter is required for successful correlation generation, which can be further improved by employing higher-precision spectral filters or different selection methods such as those discussed below and in Supplementary Section VII. Notably, the signal displayed in the correlation tomography (Fig. 4c) has been background-subtracted to reflect the net signal used in the SNR calculation Eq. (1). As shown in Eqs. (2), (3), and (4), the signal remains nearly constant across different configurations due to low crosstalk. In contrast, the background counts depicted in Fig. 4d are comparable between the correlated and uncorrelated elements, indicating that the total noise decreases approximately proportionally to the number of channels ($n$), owing to the reduced number of contributing elements in the noise term.

Based on the above processing, the SNR of rangefinding systems with varying channel numbers ($n \in 1, 2, 3$) was measured with one-second integration under different levels of solar background during daytime, as demonstrated in Fig. 4b. The experimental data, marked as symbols with different colors, were fitted to our theoretical model (Supplementary Section II), with colored curves representing different channel numbers ($n$). The agreement between the model and field trial results demonstrates the validity of the theoretical framework. The SNR improves with an increasing number of channels, particularly under higher background rates. A more pronounced noise reduction can be realized by extending the channel quantity to 80, as indicated by the purple theoretical dashed curve, with an enhancement of 5.7 dB at the maximum detected solar background level of 565 kcounts/s. Practical realization can be achieved through large-scale photonic integration combined with off-chip commercial dense wavelength division multiplexing (DWDM) components and detector array technology (Supplementary Section VII).

Experimental results under various weather conditions conducted on the same day are presented in Fig. 5. Figure 5a–d display photos of the WMB taken from Queens Building Balcony (QBB), along with the corresponding weather conditions and detected total background counts from three single-mode fiber coupled single-photon detectors. Figure 5e–h show the return photon histograms for each channel, integrated over 100 ms and displayed within a period of 20 ns. Using the previously described coincidence-based cross-correlation method, the range peaks can be identified in Fig. 5i–l. During nighttime, the echo photon peak is notably clear, with an extremely low noise level in Fig. 5e, corresponding to a distinct range peak centered at 154.81822(5) m (Fig. 5i). Although the building appears colorful at night in Fig. 5a due to visible light illumination, its influence on noise counts is negligible, as the visible light is filtered out by a long-pass filter and cannot be detected by the InGaAs single-photon detectors. During rainy moments, the detected background level increased to 8.5 kcounts/s, resulting in a dark appearance of the building in Fig. 5b. Consequently, the noise level in the photon counting histogram (Fig. 5f) rises, while the signal peak slightly decreases due to higher

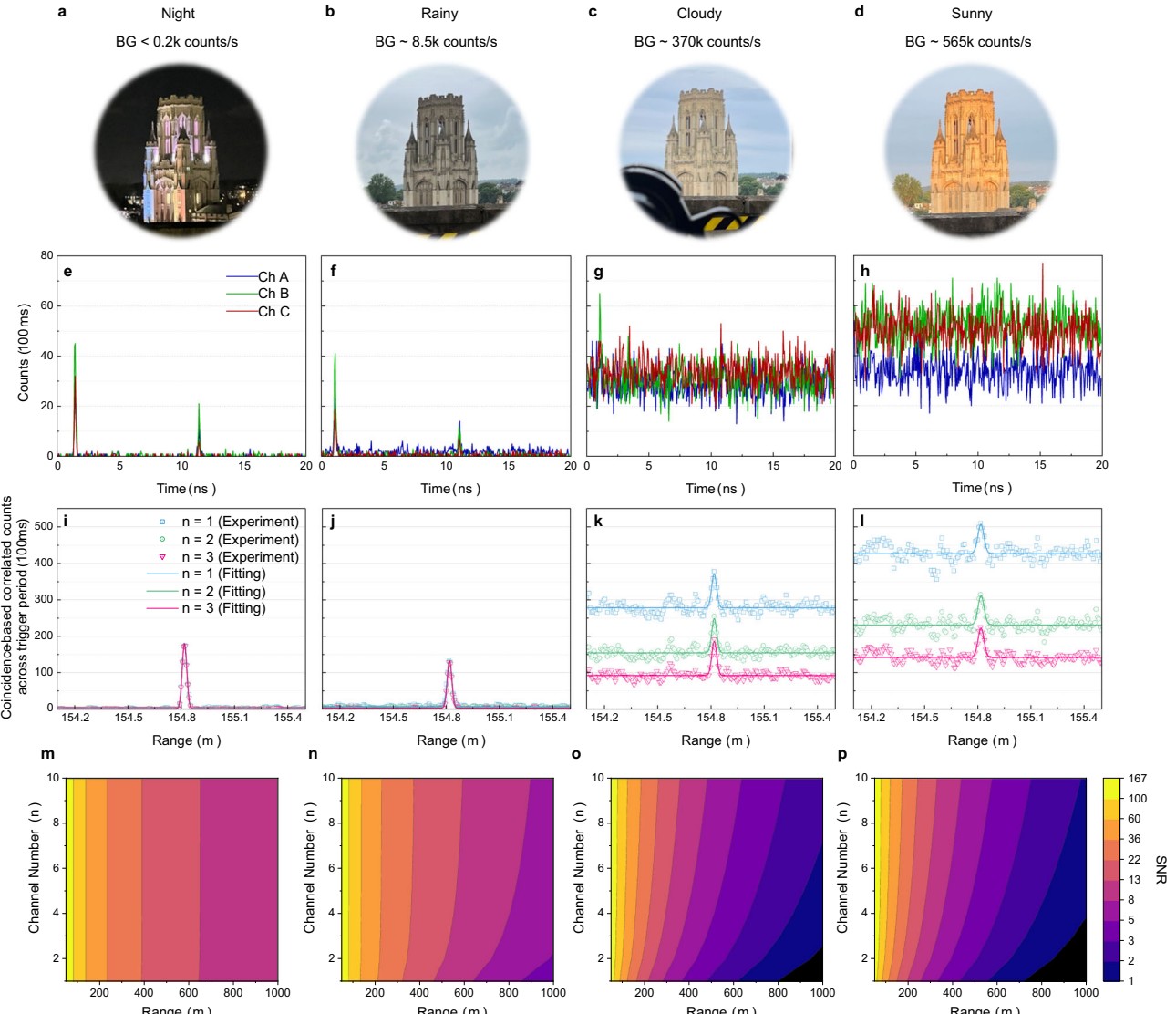

**Fig. 5 | Field trial rangefinding results. a–d** Images of the target building (WMB) under various weather conditions (BG: Total background counts measured by three single-photon detectors). **e–h** Echo photon-counting histograms for each channel (A, B, and C), displayed within single source period of 20 ns for the corresponding weather scenarios, with an integration time of 100 ms. **i–l** Coincidence-based cross-correlated peaks under different weather conditions using one channel ($n = 1$), two channels ($n = 2$), and three channels ($n = 3$), integrated over 100 ms. **m–p** Simulation results of signal-to-noise ratio (SNR) under corresponding background level as a function of channel number, $n$, and the detection range. These results demonstrate that significant noise reduction can be achieved by increasing channel numbers, particularly in strong daytime solar background.

transmission loss from the rain drops. As shown in Fig. 5j, the influence on the range peak becomes apparent, evidenced by the reduction of peak intensity. Under cloudy conditions, with measured background counts reaching 370 kcounts/s, the building appears brighter compared to the rainy period, as depicted in Fig. 5c. The background intensity in both photon-counting histogram (Fig. 5g) and correlated photon histogram (Fig. 5k) increases. However, the signal peak (after subtracting the noise level) is diminished due to the elevated background level. When sunlight illuminates the building at sunrise, as shown in Fig. 5d, the three signal peaks in the photon-counting histogram (Fig. 5h) become indistinguishable due to strong solar background noise. Nevertheless, the range peak remains detectable with the three-channel platform, while the single-channel peak becomes challenging to resolve (Fig. 5l). By comparing results from different numbers of energy channels, significant noise suppression is observed with the three-channel peak ($n = 3$), particularly under sunny conditions with higher background counts (Fig. 5l). The corresponding simulation results using the measured solar background level are

shown in Fig. 5m–p. The SNR is increased significantly with longer distance measurements due to increased transmission loss, especially under strong background levels. It should be noted that the current system can be further optimized through reductions in transmission losses and increased source power, potentially extending the detection range beyond 1 km.

## Discussion

In this work, we demonstrated the generation of energy-time correlations using a modified classical source, inspired by quantum illumination principles for noise reduction. By utilizing fiber chromatic dispersion and time-delay selection via an EOIM, multiple energy channels with corresponding time delays were generated. Consequently, energy-time correlations were successfully verified in a classical source using three isolated channels. The developed source achieves a brightness enhancement of more than 60 dB relative to conventional quantum sources, effectively overcoming the distance limitations typically associated with such systems, and can be further

enhanced easily. For instance, maximizing the extinction ratio of the EOIM by increasing the electrical signal amplitude, combined with the integration of an optical amplifier, could provide a gain of up to 23 dB. System losses can also be reduced by employing multi-mode fiber collection in conjunction with a larger-aperture telescope. The brightness enhancement extends the detection range to a Lambertian target (the external wall of the WMB), from several meters in a laboratory setting to over 100 m in outdoor environments. Experimental results demonstrate that the multi-channel ($n = 2$, $n = 3$) correlated source effectively suppresses noise relative to a direct rangefinding system ($n = 1$), with greater noise suppression achieved as the number of channels increases and negligible effect on timing jitter. Our theoretical model indicates that the noise reduction benefit seen in quantum rangefinding can be retained, and even enhanced, by increasing the number of channels, as the dark count noise remains unaffected by the addition of extra detectors. This overcomes the limitation of quantum rangefinding, which is constrained by an optimal number of channels. Field trials conducted under various daytime weather conditions including bright sunlight confirmed significant SNR improvement with the three-channel configuration, particularly in high solar background conditions, with a measured range peak at 154.81822(5) m.

In our field trials, the detection range was constrained by the available experimental space rather than the brightness of the source, indicating that further extension is possible. This potential has been demonstrated in subsequent ranging and imaging experiments, with preliminary results shown in Fig. 6. The new target is located 413.1 m away from our transmitter, as shown in the 3D Google Map in Fig. 6a. The distance was measured using the same commercial rangefinder used for measuring the distance to WMB with the resolution of 0.1 m. The return photon histograms for each channel (integrated over one second and displayed within a 20 ns window in Fig. 6b) and the correlated range peaks using multiple channels ($n = 1$, 2, and 3 in c) were detected at a distance of 413.1151 m under bright solar background conditions, demonstrating the system's potential for extending the detection range during daytime. Consistent with previous results in Fig. 5h and l, the range peak using three channels remains clearly distinguishable, even when the single-channel signal is obscured by solar background noise. Further improvements can be achieved by the incorporation of DWDM components with 80 channels which could significantly improve the SNR, offering a promising avenue for enhancing system performance under challenging environmental conditions.

The developed source also holds potential for covert rangefinding, leveraging another key advantage of quantum illumination. For instance, in terms of brightness, which is the fundamental prerequisite for all measurements, the transmission power is set to a low level of 48 μW, with pulse compression applied to obscure the generated correlations. As shown in Figs. 5h and 6b, the optical signal peaks are fully concealed within the ambient background noise, with the solar background level nearly three orders of magnitude higher than the signal power, as quantified in Supplementary Section V. To achieve fully covert operation, the unique signature of the laser source requires further manipulation. For example, the detectable repetition rate can be randomized by introducing random time delays, or being triggered by a true random sequence derived from, for example, shot noise[38], spontaneous Raman scattering[39], chaos laser[40], or a quantum random number generator[41]. Channel indices can also be encoded using a true random sequences. Spectral concealment can be further enhanced by extending the current 60 nm broadband operation to a wider spectral range through additional channels. Moreover, the laser pulses can be randomly intensity-modulated[42] or replaced with a triggered thermal source to replicate the super-Poissonian statistics of solar background. These modifications would significantly enhance covertness and reduce the likelihood of detection. Similarly the low

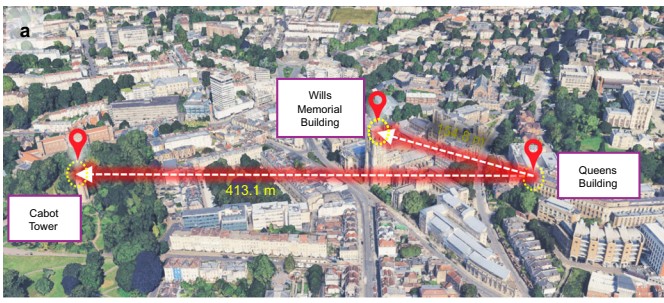

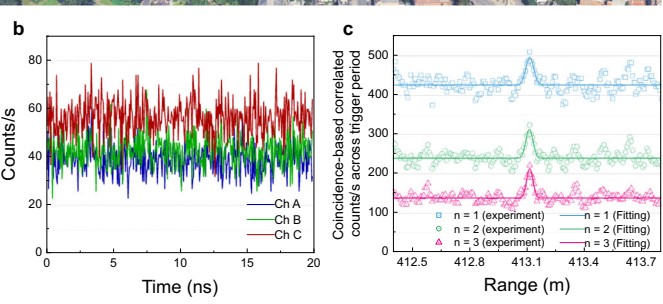

**Fig. 6 | Extended rangefinding field trial to over 400 m. a** A 3D Google map shows the distances between the balcony (Queen's Building) and the targets (both Wills Memorial Building and Cabot Tower). Underlying map from Google Earth, captured on 19 July 2024 (Data: Airbus, SIO, NOAA, U.S. Navy, NGA, GEBCO; © Google). **b** Return photon-counting histograms for each channel (A, B, and C) from Cabot Tower triggered at the source repetition rate of 50 MHz, displayed within one source period of 20 ns with one second integration time. **c** Coincidence-based cross-correlated peaks using $n$ channels ($n = 1$, 2 and 3, respectively), with an integration time of one second. The highest peak at 413.1151 m marks the measured range, fitted with a Gaussian function. Underlying map in **a** from Google Earth.

powers needed coupled with pseudo random coding could provide immunity to cross-talk in automotive Lidar applications.

## Methods
### Entanglement-inspired source
A schematic of our system is depicted in Fig. 1a. We used a fiber laser to create 58 fs long pulses at a repetition rate of 100 MHz with a central wavelength of 1563 nm and a spectral width of 67 nm. The power was carefully controlled using a fiber-coupled digital variable attenuator. The short pulses were temporally stretched to 1.2 ns using a 1 km long telecommunication band single-mode fiber (SMF28). After passing through a fiber polarization controller, the pulse was directed to an EOIM for energy channel selection. Based on the fiber chromatic dispersion, three wavelength channels were picked up within 250 ps time windows distributed over the 1.2 ns pulse duration, allowing for precise channel selection via different time delays processed by a field-programmable gate array (FPGA). The electrical signal processing is detailed in Supplementary Section III. By encoding a pseudo-random pattern to the time delay in the electrical signal, wavelength channels were randomly selected. Using this method, energy-time correlations were successfully generated from a classical source. The pulse was then split by a 1 × 2 fiber coupler with a coupling ratio of 50:50, with one path directed to a grating-based compressor used to erase the channel-selection information for covert sensing and improving distance measurement resolution. The other path was either detected by a fiber-coupled InGaAs photodiode (Thorlabs DET08CFC) connected to a fast oscilloscope (KEYSIGHT DSO-S 404A) for real-time performance monitoring, or sent directly to an optical spectrum analyser (OSA) for spectral analysis.

### Field trail of entanglement-inspired rangefinding
By integrating the developed source into a rangefinding platform, the signal was transmitted through a single-mode fiber to a collimator

positioned on the QBB, as depicted in Fig. 1b. The average transmitter power was attenuated to 48 µW for the field trial. The system had a full beam divergence of 0.032˚ (0.56 mrad), and the signal illuminated the external wall of WMB, approximately 154.8 m away from QBB, as shown in Fig. 1c. The measured distance, shown in Fig. 1d, was determined using a commercial rangefinder with a resolution of 0.1 m. The backscattered light from the building was collected by a 48 mm telescope and coupled into a single-mode fiber, passing through a long-pass filter at 1500 nm to block visible and near-infrared solar background. This filtered signal was split by a wavelength division multiplexer (WDM) and directed into three infrared single-photon detectors (ID230) through single-mode fibers, as illustrated in Fig. 1a. These InGaAs/InP photon detectors, set with a 2 µs dead time and 15% quantum efficiency, sent their detected signals to a Time Tagger (Picoquant HydraHarp 400), synchronized with the beginning of the repeating pattern down-converted by the FPGA. The field trial ran continuously from night through to day under varying weather conditions, enabling us to perform a comparison of the experimental results with our noise-reduction theory.

## Data availability

The raw and processed data generated in this study have been deposited in the Zenodo repository and are available at https://doi.org/10.5281/zenodo.18007719. Any additional materials and data are available from the corresponding author upon request.

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

## Acknowledgements

The authors acknowledge support from EPSRC through the QuantIC (EP/T00097X/1, J.G.R.), the Quantum Position, Navigation and Timing (QEPNT) Hub (EP/Z533178/1, A.S.C.), and the Quantum Sensing, Imaging and Timing (QuSIT) Hub (EP/Z533166/1, A.S.C.). A.S.C. also acknowledges support from The Royal Society (URF/R/221019, RF/ERE/210098, RF/ERE/221060).

## Author contributions

W.N., A.M., and J.G.R. conceived and designed the study. W.N. and P.Z. performed the experiments and collected the data. P.Z. programmed the FPGA for electrical signal processing. W.N., A.S.C., and J.G.R. developed the theoretical model. W.N. carried out the data analysis and numerical simulations, and wrote the manuscript with input from all authors. A.S.C. and J.G.R. reviewed and edited the manuscript. All authors discussed the results and approved the final manuscript.

## Competing interests

The authors declare no competing interests.
