## [Transparent Peer Review file · Nature Communications]

Entanglement-inspired frequency-agile rangefinding

Corresponding Author: Dr Weijie Nie

Version 0:

Reviewer comments:

Reviewer #1

(Remarks to the Author)

The authors have reported a LIDAR system enabled by time-energy correlations engineered in a modulated classical pulse. Unlike time-energy correlations in biphotons generated through nonlinear processes, the current approach is not limited by the poor efficiency of nonlinear processes or the need to operate in the low-gain regime to maintain these correlations. The authors show that using an n-channel coincidence detection scheme, the noise rejection increases with increasing number of channels n, leading to an improved signal-to-noise ratio. The authors demonstrate range measurements between two buildings, separated by 154 and 413 m, with an accuracy of 0.1 mm and a measurement time of 0.1 s.

The work is very well written and well presented. The conclusions are sound, and the arguments are very well laid out. I have a few questions and comments for the authors.

- What is the relation between the randomness of the modulation sequence and the detection distance? I didn't understand the link in line 226.

- In Figure 4, the shading in the signal box makes it impossible to see that there are 3 peaks. I recommend using an unfilled box to highlight this region. Similarly, the noise box colour blue is the same as the trace for C_AC and should be changed.

- I did not understand the improvement factor of 10^6 compared to quantum sources. The authors should clarify how they arrive at this number.

- The numbers in lines 453-468 seem a little random. While I appreciate that these are the experimental parameters, it would help the reader to understand whether these are chosen for a specific purpose or imposed by equipment limitations. This can be done in a Supplementary document.

- How would this technique compare to a direct range-finding approach using the output from the laser (pulse duration, repetition rate at the same average power) and a single detector? This would allow for a much narrower coincidence window, reducing the background and the dark counts. Would that be equivalent to the n=1 case?

Overall, I would recommend the manuscript for publication as it demonstrates clear novelty and enhanced capabilities in quantum-inspired sensing.

Bienvenu Ndagano

Reviewer #2

(Remarks to the Author)

This paper introduces a method for long-range, low-power optical rangefinding that leverages classical laser systems to emulate the noise-resilience typically associated with quantum illumination. The approach exploits energy-time correlations generated via fiber-induced chromatic dispersion and electro-optic intensity modulation. The authors demonstrate sub-millimeter ranging precision at distances exceeding 150 meters, all while operating under low transmission power, with promising potential for scalability in both range and performance.

However, in my view, the manuscript does not meet the significance threshold required for publication in Nature Communications. The novelty and impact of the work are limited when considered in the context of the current state of the field and the existing literature. A more technically focused journal may provide a more appropriate venue for this work.

General Comments:

1. A central conceptual pillar of this work is its framing as "quantum-inspired." The authors assert that the system inherits the noise-reduction benefits of quantum illumination — particularly those coming from energy-time entangled photon pairs — without relying on actual entanglement. This distinction, however, is blurred throughout the manuscript.
2. While it is true that structured correlations in time and frequency can mimic certain behaviors seen in quantum systems, the correlation mechanism here is entirely deterministic and rooted in classical signal engineering. The authors do not sufficiently delineate the theoretical boundary between what is "quantum-like" in behavior and what is simply classical modulation and post-processing. Without any true non-classical light source or statistics being involved, referring to the method as "quantum-inspired" risks overstatement.
3. One important question that arises is whether the classical correlations employed here replicate the information-theoretic and physical properties (such as resilience to loss, indistinguishability under thermal noise, and robustness against entropic decoherence) that define true quantum illumination. The paper does not explore the theoretical limits of its system in comparison to, for instance, the optimal bounds predicted by quantum Fisher information theory or Holevo capacities. As such, while the system borrows intuitions from quantum mechanics, it is not clear to what extent those benefits are fundamental versus incidental.
4. The authors derive and experimentally validate a SNR model in which dark and background noise contributions are suppressed by increasing the number of independent energy-time channels. While the data presented does support a trend of noise suppression, the analytical model assumes ideal separation between channels and uncorrelated noise across them. In real systems, however, factors such as timing jitter, spectral bleed, channel crosstalk, and detector afterpulsing can erode the orthogonality and independence assumptions underpinning this model.
5. I am a bit confused about the measurement. The experimental implementation uses three channels, the authors extrapolate their model to predict substantial benefits at 80 channels using commercial DWDM components. No information is provided about how increasing channel count affects the complexity, calibration difficulty, or cross-channel synchronization of the system. Practical deployment of an 80-channel version would face significant scaling challenges in terms of detector number, optical filtering precision, and FPGA control logic. Thus, while the observed noise suppression is genuine within the limited configuration tested, the scalability argument needs more substantiation — particularly given that real-world LIDAR applications demand robust and cost-efficient designs.
6. The authors report high range precision — down to $\pm 48 \mu\text{m}$ — by performing Gaussian fits to the correlation peak in aggregated histograms. While this is technically accurate, it is important to distinguish between measurement precision (how consistently one can locate the peak centroid) and resolution (how well two targets at different depths can be resolved). The FWHM of the system's impulse response, which defines depth resolution in practical terms, remains on the order of 25 mm.
7. One of the major weaknesses of the manuscript is the lack of systematic benchmarking against state-of-the-art classical technologies. The only substantive comparisons are made with quantum rangefinding systems based on SPDC, which are inherently limited by low brightness and high experimental fragility, especially in daylight. The high-performance LIDAR systems based on time-correlated single-photon counting, frequency-modulated continuous wave principles, and pulsed illumination with GHz detection bandwidths have made significant progress in long-range detection, noise rejection, and covert operation. Many of these use sophisticated coding schemes and multi-channel approaches not dissimilar in spirit to what is proposed here.
8. The paper makes a compelling case that the use of ultra-low transmission power and pseudo-random coding enables covert rangefinding. While the low-power operation is evident, the claim of covertness is only speculative. No data or analysis is provided to show how detectable (or undetectable) this system would be in practice against standard detection tools like optical time-domain reflectometers or RF/optical side-channel analyzers.
9. The experimental demonstration is undoubtedly impressive, particularly the ability to measure distances with sub-mm accuracy across a 150–400 meter baseline in varying weather and lighting conditions. However, the paper will benefit from additional discussion of environmental robustness — for example, whether alignment drift, thermal effects, or fiber movement degrade performance. Additionally, long-term stability tests and repeatability of precision across independent trials are not discussed.

Version 1:

Reviewer comments:

Reviewer #1

(Remarks to the Author)

The authors have satisfactorily replied to my questions and comments

Reviewer #2

(Remarks to the Author)

The authors have addressed my questions and revised the manuscript appropriately. I therefore recommend the publication of the work in Nature Communications.

Response to Reviewers' Comments:

Reviewer #1 (Remarks to the Author):

The authors have reported a LIDAR system enabled by time-energy correlations engineered in a modulated classical pulse. Unlike time-energy correlations in biphotons generated through nonlinear processes, the current approach is not limited by the poor efficiency of nonlinear processes or the need to operate in the low-gain regime to maintain these correlations. The authors show that using an n-channel coincidence detection scheme, the noise rejection increases with increasing number of channels n, leading to an improved signal-to-noise ratio. The authors demonstrate range measurements between two buildings, separated by 154 and 413 m, with an accuracy of 0.1 mm and a measurement time of 0.1 s.

The work is very well written and well presented. The conclusions are sound, and the arguments are very well laid out. I have a few questions and comments for the authors.

We sincerely thank the reviewer for the positive comments on our manuscript, as well as for the thoughtful questions and constructive suggestions, which help us to further clarify and strengthen the work. Our detailed responses to each point are provided below.

1.1 What is the relation between the randomness of the modulation sequence and the detection distance? I didn't understand the link in line 226.

Response:

The detection distance is inherently limited by the repetition rate when a periodic pulsed laser is used as the source in a time-of-flight measurement. The distances can only be determined unambiguously when the pulse repetition rate is sufficiently low to ensure that only one pulse is in transit at any given time, limiting the source brightness. In other words, if the return signal from a distant target arrives after the next pulse has been emitted, the system cannot determine which emitted pulse it corresponds to, leading to range ambiguity. The maximum distance at which pulses can be uniquely identified, known as the unambiguous range, is given by [1]

$$R_{unambiguous} = \frac{c}{2 \cdot f}$$

where f is the laser repetition rate and c is the speed of light. In our system, the original laser operates at a repetition rate of **50 MHz** (after processing by the FPGA limited by the time-delay module), corresponding to an unambiguous range of only **3 m**.

To overcome this limitation, we employ a pseudo-random channel sequence (PRCS) of length **500** by applying fibre-induced chromatic dispersion and pulse carving for wavelength-channel encoding in the developed source. This encoding effectively increases the unambiguous range

to **1500 m**, as the return signals can be uniquely identified based on their wavelength-dependent timing positions. Consequently, this approach extends the unambiguous detection distance while maintaining accurate signal recognition. For clarity, we have revised the corresponding sentences in the manuscript and included an additional reference, as shown below.

The revised version:

Lines 236 - 240: This breaks the limit of a periodic source on maximum **unambiguous** detection distance [1], caused by the high repetition rate of 50 MHz, and could be further extended using **a longer random sequence or even** a real-random sequence.

[1] Krichel, N. J., McCarthy, A. & Buller, G. S. Resolving range ambiguity in a photon counting depth imager operating at kilometer distances. *Opt. Express* **18**, 9192-9206 (2010).

1.2 In Figure 4, the shading in the signal box makes it impossible to see that there are 3 peaks. I recommend using an unfilled box to highlight this region. Similarly, the noise box colour blue is the same as the trace for C_AC and should be changed.

Response:

We thank the reviewer for this helpful suggestion. We have removed the shading in both the signal and noise boxes to improve visibility in Fig. 4(a). To avoid colour overlap, the noise box dashed outline has been changed to a dark blue colour (Fig.4(a)). In addition, we have replaced the word “noise” with “background” in Figs. 4(a) and (c) to avoid confusion between noise and solar background, where noise refers to fluctuations and background denotes solar photon counts collected by the single-photon detectors.

The revised version: The revised figure is shown below for clarity.

1.3 I did not understand the improvement factor of 10^6 compared to quantum sources. The authors should clarify how they arrive at this number.

Response:

We appreciate the reviewer's careful evaluation and the emphasis on the improvement factor, which refers to the brightness enhancement of the developed source. The quantitative improvement is described in the Results section, where we state:

"The quantum-inspired correlation source achieves an average power of 147.5 μ W before the fibre splitter, corresponding to 23 million photons per pulse. This represents more than 60 dB improvement compared to conventional quantum sources that are forced to operate at one photon per time window on average."

The brightness of the quantum-inspired source is calculated from its achievable average power of 147.5 μ W, corresponding to approximately 1.15×10^{15} photons per second. In comparison, a conventional quantum source operating 0.1 pairs per time window (normally > 100 ps) has a max brightness of 1×10^9 photons per second. This demonstrates an enhancement of more than 60 dB in brightness. To improve clarity, we have revised the content in lines 163-170 and also replaced the improvement factor to 60 dB in Discussion (line 378-382).

The revised version:

1. Lines 163-170: The quantum-inspired correlation source **can** achieve an average power of 147.5 μ W before the fibre splitter, **corresponding to 1.15×10^{15} photons per second**. This represents more than 60 dB improvement compared to conventional quantum sources **of 1×10^9 photons per second assuming 0.1 photon pairs per time window (100 ps) on average**.
2. Lines 378-382: **The developed source achieves a brightness enhancement of more than 60 dB relative to conventional quantum sources, ...**

1.4 The numbers in lines 453-468 seem a little random. While I appreciate that these are experimental parameters, it would help the reader to understand whether these are chosen for a specific purpose or imposed by equipment limitations. This can be done in a Supplementary document.

Response:

We appreciate the reviewer's suggestion. The relevant content has been revised to better explain the selection purpose and relocated to the Supplementary Section III.

The revised version:

1. Add a sentence in the manuscript (lines 486-487): **The electrical signal processing is detailed in Supplementary Section III.**
2. Original lines 501-517 in main manuscript are relocated to **Supplementary information Section III:**

Electrical Signal Processing: To realize precise energy channel selection, the EOIM is driven by a 250-ps electrical signal with a variable time delay programmed by the FPGA (Xilinx ZYNQ-7000). The original electrical pulse is derived from the femtosecond laser to ensure synchronization, featuring a pulse width of 370 ps and a repetition rate of 100 MHz. The pulses are subsequently broadened to 4 ns by a pulse shaper to be reliably detected by the FPGA input, and are then down-converted to 50 MHz, as constrained by the operating range of the FPGA's IDELAY module. A pseudo-random sequence of variable time delays is generated and encoded by the internal IDELAY module onto 500 consecutive pulses, synchronized with the down-converted 100-kHz reference signal sent to the time-tagger (HydraHarp 400). Finally, the electrical pulses are compressed to 250 ps (to enhance temporal resolution) and amplified to 3 V (to improve the extinction ratio, though still below the full 6 V drive limit due to amplifier saturation). These pulses are then applied to the EOIM, which serves as a high-speed optical switch that randomly selects wavelength channels according to the encoded pattern.

1.5 How would this technique compare to a direct range-finding approach using the output from the laser (pulse duration, repetition rate at the same average power) and a single detector? This would allow for a much narrower coincidence window, reducing the background and the dark counts. Would that be equivalent to the $n=1$ case?

Response:

We thank the reviewer for this insightful question, which highlights the key aspect of our work. To demonstrate the noise-reduction advantage, we experimentally compared our method with a direct rangefinding approach corresponding to the $n = 1$ case with the same detector, same average source power and repetition rate, same pulse duration, as well as the same on/off ratio of the pseudo-random pattern used to avoid the 3-metre unambiguous-range limitation of the 50-MHz output laser.

Regarding the coincidence window, it remains identical to that used in the multi-channel system since the laser pulse duration and detector timing jitter are the same. Therefore, the background and dark counts cannot be reduced by coincidence gating alone. In contrast, the background level increases because the collected photon counts are not filtered through energy-time correlations. In other words, uncorrelated photon counts are fully detected,

leading to a significant rise in background noise as illustrated in Fig. 4, where all uncorrelated photon counts are included in the direct rangefinding case.

We appreciate the reviewer's comment and have added a clarifying sentence in the revised manuscript to better connect the concept of direct rangefinding with the $n = 1$ case.

Thus, we have proved our technique works with the present pulse width and detector jitter without broadening the coincidence window significantly. If we were to implement it with better detectors and shorter pulses, an $n = 3$ system would still suppress background noise compared to the $n = 1$ system.

The revised version:

Lines 394-399: Experimental results demonstrate that the multi-channel ($n = 2$, $n = 3$) correlated source effectively suppresses noise relative to a direct rangefinding system ($n = 1$), with greater noise suppression achieved as the number of channels increases and negligible effect on timing jitter.

Overall, I would recommend the manuscript for publication as it demonstrates clear novelty and enhanced capabilities in quantum-inspired sensing.

Response: Thank you again for the highly positive comments. We have carefully addressed all the questions and suggestions, which have helped us further refine the presentation and enhance the overall quality of the manuscript.

Reviewer #2 (Remarks to the Author):

This paper introduces a method for long-range, low-power optical rangefinding that leverages classical laser systems to emulate the noise-resilience typically associated with quantum illumination. The approach exploits energy-time correlations generated via fiber-induced chromatic dispersion and electro-optic intensity modulation. The authors demonstrate sub-millimeter ranging precision at distances exceeding 150 meters, all while operating under low transmission power, with promising potential for scalability in both range and performance.

However, in my view, the manuscript does not meet the significance threshold required for publication in Nature Communications. The novelty and impact of the work are limited when considered in the context of the current state of the field and the existing literature. A more technically focused journal may provide a more appropriate venue for this work.

Response:

We appreciate the reviewer's feedback on the revised manuscript. However, we respectfully disagree with the assessment regarding the novelty and impact of our work. The presented results establish a new framework for quantum-inspired optical rangefinding, bridging quantum concepts with practical remote-sensing technologies. The main highlights are as follows:

- 1. Translation of quantum noise-reduction to a classical platform:** Our quantum-inspired rangefinding system successfully translates the core noise-reduction mechanism of quantum lidar into a classical architecture, as verified through both theoretical analysis and experimental validation. The noise-suppression effect is preserved and even enhanced by increasing the number of correlation channels, since dark-count noise remains largely unaffected by the addition of detectors. This overcomes a key limitation of quantum rangefinding, which is restricted by an optimal channel number.
- 2. Significant improvement in source brightness and operational range:** The source brightness is increased by more than 60 dB compared to quantum sources, effectively overcoming the range limitation that has so far confined demonstrations of noise-reduction advantages to short laboratory distances. Conventional quantum rangefinding typically operates over only a few metres, making it difficult to observe significant noise-resilience effects and limiting applicability in real-world conditions. Our approach removes this constraint, enabling practical deployment while maintaining the noise-reduction advantages of quantum rangefinding. This represents the first real-world demonstration of a quantum-inspired rangefinding system operating beyond laboratory settings.
- 3. Demonstrated noise-resilience and robustness in real-world field trials:** Successful field trials under challenging environmental conditions, including ultra-low source power (48 μ W), extremely high background (direct solar irradiation), long ranges (155 m and 413 m), and a non-cooperative target (a building facade), demonstrate the

practical noise-resilience and robustness of the proposed approach. Even under noise-dominated conditions (Fig. 5(d) and Fig. 6), our system can still retrieve range information from indistinguishable signal photons. Furthermore, its stable performance under varying weather and solar illumination confirms its readiness for real-world sensing applications.

A detailed explanation and a comprehensive comparison with state-of-the-art have been provided in **Q2.7** and added to the **Supplementary Information** for clarity.

General Comments:

- 2.1 A central conceptual pillar of this work is its framing as "quantum-inspired." The authors assert that the system inherits the noise-reduction benefits of quantum illumination — particularly those coming from energy-time entangled photon pairs — without relying on actual entanglement. This distinction, however, is blurred throughout the manuscript.
- 2.2 While it is true that structured correlations in time and frequency can mimic certain behaviors seen in quantum systems, the correlation mechanism here is entirely deterministic and rooted in classical signal engineering. The authors do not sufficiently delineate the theoretical boundary between what is "quantum-like" in behavior and what is simply classical modulation and post-processing. Without any true non-classical light source or statistics being involved, referring to the method as "quantum-inspired" risks overstatement.
- 2.3 One important question that arises is whether the classical correlations employed here replicate the information-theoretic and physical properties (such as resilience to loss, indistinguishability under thermal noise, and robustness against entropic decoherence) that define true quantum illumination. The paper does not explore the theoretical limits of its system in comparison to, for instance, the optimal bounds predicted by quantum Fisher information theory or Holevo capacities. As such, while the system borrows intuitions from quantum mechanics, it is not clear to what extent those benefits are fundamental versus incidental.

Response:

We thank the reviewer for raising these three comments regarding the distinction between **quantum** and **quantum-inspired** rangefinding, and we address them together below.

For remote detection under high transmission loss and strong background noise, the noise-reduction advantage observed in quantum rangefinding primarily arises from energy-time correlations across multiple frequency channels [1,2]. This correlation structure can persist even after entanglement is degraded, as discussed by Lloyd [2]. In this sense, the substantial

enhancement in noise resilience associated with quantum illumination is rooted in the correlations themselves, rather than in the presence of entanglement. By contrast, the benefit uniquely attributable to entanglement has been shown to be limited to a 6 dB improvement and requires operation near the time-bandwidth limit with very low photon numbers per timing window. This inherently limits the achievable source brightness by several orders of magnitude compared to our system and therefore restricts quantum rangefinding to short distances and highly reflective targets such as mirrors [3,4].

Our work focuses on reproducing correlation-based enhancement using a classical optical source suitable for realistic remote sensing scenarios, thereby achieving comparable noise suppression without relying on entanglement. In our system, the source operates with a pulse duration exceeding 180 ps and a spectral bandwidth above 8 nm, corresponding to a time-bandwidth product much greater than unity. Therefore, the observed noise-reduction effect arises entirely from energy-time correlations which is exactly the case in recent quantum rangefinder demonstrations [1], but in our case the correlations are classically engineered.

- 2.1. In our terminology, **quantum rangefinding** refers to systems based on entangled-photon sources, whereas **quantum-inspired rangefinding** refers to systems that employ classically generated correlations. We have clarified this distinction explicitly in the **Introduction**.
- 2.2. The term **quantum-inspired** is used intentionally to indicate that our approach is conceptually derived from the principles of quantum rangefinding. While the system does not employ non-classical light or entanglement, it retains the primary quantum-like advantage of enhanced noise suppression, rather than quantum-specific behaviour such as entanglement itself.
- 2.3. Our objective is not to recreate entanglement in the source. Instead, we replicate the primary operational noise-suppression benefit of quantum rangefinding by implementing engineered classical correlations.
 - a. **Resilience to loss:** The developed source is more tolerant to high transmission loss than entangled sources, which aligns with our experimental results under varying weather conditions, detection distances, and continuous 24-hour operation (Fig. 5, Fig. 6, and Fig. S2).
 - b. **Indistinguishability under thermal noise:** In our experiments, the solar background exhibits near-Poissonian statistics because the detector time-bandwidth product is larger than unity (timing jitter ~ 150 ps and bandwidth ~ 10 nm). Under these detection conditions, our source is indistinguishable from the solar background. Demonstrating thermal photon statistics would require operating at a time-bandwidth product near unity, which is not applicable to our system.
 - c. **Robustness against decoherence:** Our system does not rely on phase coherence (e.g., narrowband interference). As a result, the source remains

robust, and the original laser source could in principle be replaced with a thermal source without altering the performance.

For these reasons, analyses based on quantum Fisher information or Holevo capacities are not informative, as the system operates entirely in the classical regime, and such bounds would not help optimize our approach.

The revised version:

Lines 108-115: This quantum-inspired rangefinding employs classically correlated optical sources while preserving most of the noise-reduction benefits of quantum rangefinding, which uses entangled-photon sources. To achieve the benefit of entanglement requires operating at the time-bandwidth limit at low photon numbers per timing window, limiting operational distance [3].

[1] Frick, S., McMillan, A. & Rarity, J. Quantum rangefinding. *Opt. Express* **28**, 37118-37128, (2020).

[2] Lloyd, S. Enhanced Sensitivity of Photodetection via Quantum Illumination. *Science* **321**, 1463-1465, (2008).

[3] Tan, S.-H. *et al.* Quantum Illumination with Gaussian States. *Phys. Rev. Lett.* 101, 253601, (2008).

[4] Shapiro, J. H. & Lloyd, S. Quantum illumination versus coherent-state target detection. *New J. Phys.* **11**, 063045, (2009).

2.4. The authors derive and experimentally validate a SNR model in which dark and background noise contributions are suppressed by increasing the number of independent energy-time channels. While the data presented does support a trend of noise suppression, the analytical model assumes ideal separation between channels and uncorrelated noise across them. In real systems, however, factors such as timing jitter, spectral bleed, channel crosstalk, and detector afterpulsing can erode the orthogonality and independence assumptions underpinning this model.

Response:

We thank the reviewer for raising this important point, which has been considered in the Supplementary Information (Lines 60–61):

“ η_i is the unbalance in channel power splitting with $\sum_{i=1}^n \eta_i = n$,”

The non-ideal separation between different channels has been incorporated into the SNR model through an imbalance parameter η_i . As shown in the detailed derivation, this non-ideal

separation does not affect the final SNR expression. Specifically, when substituting Eq. (S6) into the signal term in Eq. (S5), we obtain

$$\sum_{i=1}^n C_{Si} = \sum_{i=1}^n \frac{N_S}{n} \eta_i \eta_S = N_S \eta_S \sum_{i=1}^n \eta_i = N_S \eta_S$$

which demonstrates that the imbalance parameters cancel out when all channels are summed.

For uncorrelated noise, we initially did not include imbalance parameters, as their contribution is negligible under strong background conditions. To make the model more rigorous, we have now introduced imbalance parameters for both background and dark count terms. This refinement aligns with our experimental results, where three channels with inherent imbalance showed consistent agreement with the theoretical model.

Regarding the reviewer's concern about practical non-idealities such as timing jitter, spectral bleed, channel crosstalk, and detector afterpulsing, we agree that maintaining channel orthogonality and independence is essential for preserving the generated energy-time correlations. These aspects are discussed and experimentally verified in our manuscript. Specifically, we characterized the source in both the time and energy domains (Fig. 2), showing low spectral bleed (>10dB isolation between channels). Furthermore, as shown in Fig. 4(c), inter-channel crosstalk contributes to the matrix elements C_{ij} where $i \neq j$, which partially explains the small discrepancy between experimental and theoretical results in Fig. 4(b). This effect can be further minimized by employing higher-precision spectral filters in the source or using different methods for energy-time correlation generation.

To ensure completeness, we have refined the model by incorporating imbalance parameters for the noise terms and added a discussion on addressing the influence of practical non-idealities.

The revised version

Supplementary Information:

1. Equation S8 and Lines 73-75: η_{ub} is the channel unbalance in background counts splitting with $\sum_{ub=1}^n \eta_{ub} = 1$;
2. Equation S9 and Lines 88-90: η_{ud} is the imbalance in the dark counts of different detectors related to the channel splitting, where $\sum_{ud=1}^n \eta_{ud} = 1$;
3. Lines 101-102: Upon summation over all channels, all imbalance parameters (η_i , η_{ub} and η_{ud}) are effectively cancelled.

Main manuscript:

4. Lines 273-282: Uncorrelated counts ($i \neq j$), which are primarily induced by the channel crosstalk and time jitter, can be minimized by increasing the delay between adjacent channels. As channel orthogonality and independence are crucial for the developed

energy-time correlations, low crosstalk and minimal timing jitter is required for successful correlation generation, which can be further improved by employing higher-precision spectral filters or different selection methods such as those discussed below and in Supplementary Section VII.

2.5. I am a bit confused about the measurement. The experimental implementation uses three channels; the authors extrapolate their model to predict substantial benefits at 80 channels using commercial DWDM components. No information is provided about how increasing channel count affects the complexity, calibration difficulty, or cross-channel synchronization of the system. Practical deployment of an 80-channel version would face significant scaling challenges in terms of detector number, optical filtering precision, and FPGA control logic.

Thus, while the observed noise suppression is genuine within the limited configuration tested, the scalability argument needs more substantiation — particularly given that real-world LIDAR applications demand robust and cost-efficient designs.

Response:

We thank the reviewer for raising the important issue of scalability. We fully agree that practical deployment requires careful consideration of system complexity, detector resources, filtering precision, and synchronization. In principle, the scalability of our approach is technically feasible:

1. The complexity and calibration challenges can be mitigated using integrated photonic platforms [1,2], which provide compact, robust, and cost-effective solutions for increasing the number of channels. Once the chip design is finalized, the chip-based configuration offers stable alignment and straightforward operation.
2. Detector requirements can be addressed by employing InGaAs detector arrays [3,4] or by operating in the near-infrared or visible regions, where high-performance silicon detectors are readily available.
3. Optical filtering precision can be achieved using mature commercial DWDM components [5,6], which routinely support 96 channels with low crosstalk (adjacent channel isolation > 25dB) and stable spectral performance. Such filtering can also be realized using integrated arrayed waveguide gratings [7], which provide compact and robust multiplexing solutions.
4. The FPGA we use is an intermediate model that is cost effective (~£230) and can be used for up to 16 channels. To increase channel number, we can synchronise multiple of these together [8] or opt for a higher specification model.

These clarifications highlight the realistic path toward robust, large-scale deployments of our quantum-inspired noise-suppression approach for rangefinding applications. Currently we are investigating an integrated 10-channel photonic chips on InP platform, which further

substantiates the feasibility of scaling to high-channel-count, cost-efficient architectures. To clarify these aspects of scalability, we added a section in the supplementary information.

The revised version:

Manuscript Lines 311-315: Practical realization can be achieved through large-scale photonic integration combined with off-chip commercial dense wavelength division multiplexing (DWDM) components and detector array technology (Supplementary Section VII).

Supplementary Information (SI): A new **SECTION VII SCALABILITY** is added in the SI:

Considering scalability to a larger number of channels, integrated photonic platforms [1,2] represent a natural and practical path forward. They effectively address system complexity and calibration challenges while offering compact, robust, and cost-efficient solutions for multi-channel optical architectures. Once the chip design is finalized, the integrated configuration provides intrinsically stable alignment and straightforward operation. Detector requirements can be addressed by employing InGaAs detector arrays [3,4] or by operating in the near-infrared or visible regions, where high-performance silicon detectors are readily available. High-precision optical filtering can be achieved using mature commercial DWDM components [5,6], which routinely support 96 channels with low crosstalk (adjacent channel isolation >25 dB) and excellent spectral stability. Alternatively, integrated arrayed waveguide gratings offer a compact and robust solution for scalable wavelength multiplexing [7]. Finally, the FPGA we use is an intermediate model that is cost effective and can be used for up to 16 channels. To increase channel number, we can synchronise multiple of these together [8] or opt for a higher specification model, enabling parallel processing and straightforward system expansion.

Ref:

[1] Xu, X., Ren, G., Feleppa, T., *et al.* Self-calibrating programmable photonic integrated circuits. *Nat. Photon.* **16**, 595-602 (2022).

[2] Pérez-López, D., Torrijos-Morán, L. Large-scale photonic processors and their applications. *npj Nanophoton.* **2**, 32 (2025).

[3] Dolphin J. A., Scowen R. O. E., Wells L. M., *et al.* Hybrid-Integrated InGaAs/InP SPAD Arrays for Quantum Communications. *arXiv preprint* arXiv:2509.05134, (2025).

[4] Tan, C. *et al.* Development of a near-infrared single-photon 3D imaging LiDAR based on 64×64 InGaAs/InP array detector and Risley-prism scanner. *Opt. Express* **32**, 7426-7447 (2024).

[5] Ohara, T. *et al.* Over-1000-channel ultradense WDM transmission with supercontinuum multicarrier source. *J. Lightw. Technol.* **24**, 2311-2317, (2006).

[6] Singh, S. & Kaler, R. S. Comparison of pre-, post- and symmetrical compensation for 96 channel DWDM system using PDCF and PSMF. *Optik* **124**, 1808-1813, (2013).

[7] Wang Z., Fang Z., Liu Z., *et al.* On-Chip Arrayed Waveguide Grating Fabricated on Thin-Film Lithium Niobate. *Adv. Photon. Res.*, 5(2), 2300228 (2024).

[8] Xu Y., Rajagopala A. D., Fruitwala N., *et al.* Multi-FPGA Synchronization and Data Communication for Quantum Control and Measurement. *arXiv preprint* arXiv:2506.09856, (2025).

2.6. The authors report high range precision — down to $\pm 48 \mu\text{m}$ — by performing Gaussian fits to the correlation peak in aggregated histograms. While this is technically accurate, it is important to distinguish between measurement precision (how consistently one can locate the peak centroid) and resolution (how well two targets at different depths can be resolved). The FWHM of the system's impulse response, which defines depth resolution in practical terms, remains on the order of 25 mm.

Response:

We thank the reviewer for this comment, which aligns with the distinction we have made in the manuscript. As correctly noted, measurement precision and resolution represent two distinct aspects of system performance. The measurement precision reflects the statistical uncertainty in locating the centroid of the correlation peak, whereas the single-shot depth resolution is determined by the full width at half maximum (FWHM) of the three-channel correlation histogram. To clarify this distinction, we have revised the relevant sentences in the manuscript as follows:

The revised version:

Lines 223-234: ... centred at a distance of 154.81822(5) m. This fitting improves the measurement precision to $\pm 48 \mu\text{m}$, representing the statistical uncertainty in locating the peak centroid and overcoming the limitations set by detector jitter. In contrast the single shot depth resolution is determined by the full width at half maximum (FWHM) of combined the three-channel histogram shown in Fig. 3(d), which is 25.7(1) mm. This value indicates how well two targets at different depths can be distinguished.

2.7. One of the major weaknesses of the manuscript is the lack of systematic benchmarking against state-of-the-art classical technologies. The only substantive comparisons are made with quantum rangefinding systems based on SPDC, which are inherently limited by low brightness and high experimental fragility, especially in daylight.

The high-performance LIDAR systems based on time-correlated single-photon counting, frequency-modulated continuous wave principles, and pulsed illumination with GHz detection bandwidths have made significant progress in long-range detection, noise rejection, and

covert operation. Many of these use sophisticated coding schemes and multi-channel approaches not dissimilar in spirit to what is proposed here.

Response:

We thank the reviewer for this valuable comment regarding benchmarking. In our work, we have included comparisons with both classical and quantum rangefinding systems. For the classical case, as also noted by the first reviewer, we defined the direct rangefinding configuration as the single-channel case ($n = 1$). This provides a fair and controlled benchmark, as all experimental conditions remain identical except for the number of channels. The observed improvement in noise suppression with increasing channel number ($n = 2$ and 3) not only validates our theoretical model but also demonstrates a direct and practical performance gain relative to classical systems. For quantum rangefinding, comparisons have been made with previously reported systems based on SPDC sources, as recognized by the second reviewer.

Additionally, we have included a comprehensive state-of-the-art performance comparison and detailed discussion in the Supplementary Information. This survey spans quantum rangefinding [1], Chaotic-QFC LiDAR (Quantum-inspired LiDAR) [2], High Performance Single Photon LiDAR [3-6], GHz-SPAD LiDAR [7], and FMCW LiDAR [8,9]. The comparison table evaluates key performance metrics, including source average power, detection range, detected target, collection aperture, detector type, and background noise, enabling an assessment of long-range capability under operationally critical conditions.

Regarding “sophisticated coding schemes and multi-channel approaches” mentioned by reviewer, sophisticated coding schemes [3,10] have traditionally been employed to extend the ambiguity range imposed by a fixed repetition rate. Multi-channel systems [6,11] have primarily been reported in spectral-scanning applications with grating, where wavelength channels are distributed spatially rather than temporally onto the illuminated target, as implemented in this work. Although the corresponding experimental configurations may appear superficially similar, the underlying principles, intended applications, and resulting performance characteristics differ substantially from those reported previously. Crucially, no prior system has combined both architectures with wavelength-time coding (i.e., energy-time correlation) to achieve noise reduction. This integration represents a key innovation of the present work, enabling long-range, low-power operation under strong and realistic background illumination, where capabilities not demonstrated in earlier studies.

The revised version:

Supplementary Information: **Section VI. STATE-OF-THE-ART COMPARISON**

A comprehensive comparison of state-of-the-art performance metrics is provided in Table S1, covering quantum rangefinding [1], chaotic quantum frequency conversion (chaotic-QFC, quantum-inspired) Light Detection and Ranging (LiDAR) [2], high-performance single-photon

LiDAR systems [3–6], Gigahertz single-photon avalanche diode (GHz-SPAD) LiDAR [7], and frequency-modulated continuous-wave (FMCW) LiDAR technologies [8,9]. Compared with quantum ranging demonstrations [1], the source brightness employed here is substantially higher yet remains within the covert-illumination limits established in the covert-analysis (Supplementary Information Section V). This enables long-distance operation without exceeding covert thresholds. By contrast, most classical LiDAR systems use optical powers > 1 mW, at least 20 times higher than in this work, and therefore operate outside the covert regime. Achieving ranges beyond 1 km with a low-reflective target typically requires either more than Watt illumination or highly sensitive, cryogenic detectors such as superconducting nanowire single-photon detectors (SNSPDs) in the reported systems. This is either not good for brightness covertness or compact detection system. It is worth noting that our approach can readily support longer distance through increased optical power or extended integration times.

Our experimental conditions also impose more stringent loss factors than those used in many previous field demonstrations. Measurements were performed using an older building with an irregular, low-reflectivity facade, introducing significant surface-scattering loss. In contrast, many long-range studies use high-reflectivity targets, where losses are dominated primarily by beam divergence. The receiver optics in our system consist of a compact 48 mm aperture, offering a portable and practical collection system for deployment. For detection, we use single-photon avalanche diodes (SPADs), which offer an effective balance of sensitivity, robustness, and cost. This contrasts with homodyne-based receivers, which, despite their sensitivity, are mechanically fragile and challenging to deploy outside the laboratory, and with SNSPDs, which are costly and require continuous cryogenic cooling.

The background illumination conditions in our measurements further represent a demanding and realistic outdoor scenario. As shown in Fig. 5(d), the target building is directly exposed to sunlight, resulting in a solar background level nearly three orders of magnitude higher than the transmitted power, as quantified in the covertness analysis in Section V. Many reported studies either simulate ambient noise in controlled laboratory settings or conduct measurements under standard daylight conditions without quantifying the background level. The results presented here therefore demonstrate system performance under overwhelmingly strong and realistic background noise levels encountered in practice.

In conclusion, our field trial employed an ultra-low-power ($48 \mu\text{W}$) source to detect a remote target directly illuminated by sunlight, representing an extremely high-background scenario. Even in these noise-dominated conditions, where the ambient background exceeded the signal by nearly three orders of magnitude, the system successfully recovered signal photons. These results highlight the practical robustness and exceptional noise tolerance of the approach, demonstrating its suitability for real-world deployment.

Table S1 State-of-the-Art Performance Comparison for Quantum and Classical Lidar System

System	Avg. Tx Power	Detection Range	Detected Target	Collection Aperture	Detector Type	Background Noise
Quantum-inspired Rangefinding (This work)	48 μ W	413 m	Building	48 mm	SMF-coupled SPAD	Direct solar irradiation on target (Background $\approx 10^3 \times$ signal)
Quantum Rangefinding [1]	pW-scale	3 m (lab)	Corner-cube	48 mm	Si APD	LED-simulated
Chaotic-QFC LiDAR (Quantum-inspired LiDAR) [2]	50 mW	4 m (lab)	Indoor quadcopter	50.8 mm	Balanced Homodyne Detection	Simulated by amplified spontaneous emission (ASE) noise
SNSPD Single Photon LiDAR [3]	2.9 mW	1 km	Comms mast (metal frame)	254 mm	Single-mode-fibre (SMF) coupled SNSPD	Daylight
Satellite Single-Photon LiDAR [4]	15 W	935.9 km	Retroreflective satellite	280 mm	SMF-coupled SPAD	Night
Single Photon LiDAR with 64x64 SPAD Array [5]	2.5 W	5.7 km	Building	50 mm	64x64 InGaAs/InP SPAD array	Daylight
High-Repetition-Rate Single Photon LiDAR [6]	200 mW	340 m	DJI mini3	40 mm	Multi-mode-fibre (MMF) coupled SNSPD	Not stated
Airborne GHz-SPAD LiDAR [7]	107 mW	620 m	Ground	15.3 mm	MMF-coupled SPAD	Morning (Background $\approx 30 \times$ signal)
OPA-based FMCW LiDAR [8]	≤ 40 W	2 km	Building	350 mm	Free space InGaAs APD	Night
Long-range FMCW LiDAR [9]	1 W CW	3 km	Houses	20 mm	Coherent balanced photodiodes	Clear air

Ref:

[1] Frick, S., McMillan, A. & Rarity, J. Quantum rangefinding. *Opt. Express* **28**, 37118-37128, (2020).

[2] Liu, H., Qin, C., Papangelakis, G., Lu, M. L. & Helmy, A. S. Compact all-fiber quantum-inspired LiDAR with over 100 dB noise rejection and single photon sensitivity. *Nat. Commun.* **14**, 5344, (2023).

[3] McCarthy, A. *et al.* High-resolution long-distance depth imaging LiDAR with ultra-low timing jitter superconducting nanowire single-photon detectors. *Optica* **12**, 168-177, 77 (2025).

[4] Li, Y. *et al.* Compact single-photon LiDAR for satellite laser ranging. *Opt. Express* **33**, 40876-40890, (2025).

[5] Tan, C. *et al.* Development of a near-infrared single-photon 3D imaging LiDAR based on 64x64 InGaAs/InP array detector and Risley-prism scanner. *Opt. Express* **32**, 7426-7447, (2024).

[6] Zhang, W. *et al.* Sensitive micro UAV detection based on a high-repetition-rate single-photon LiDAR. *Opt. Express* **33**, 18102-18111, (2025).

- [7] Shen, G. *et al.* High-speed airborne single-photon LiDAR with GHz-gated single-photon detector at 1550 nm. *Opt. Laser Technol.* **141**, 107109, (2021).
- [8] Wu, Y. *et al.* Multi-beam optical phase array for long-range LiDAR and free-space data communication. *Opt. Laser Technol.* **151**, 108027, (2022).
- [9] Fenevrou, P., Martin, A., Dolfi, D. & Payot, E. 3D imaging with large range dynamics and simultaneous accurate speed measurement. *Appl. Opt.* **63**, 5387-5394, (2024).
- [10] Krichel, N. J., McCarthy, A. & Buller, G. S. Resolving range ambiguity in a photon counting depth imager operating at kilometer distances. *Opt. Express* **18**, 9192-9206 (2010).
- [11] Jiang, Y., Karpf, S. & Jalali, B. Time-stretch LiDAR as a spectrally scanned time-of-flight ranging camera. *Nat. Photonics.* **14**, 14-18, (2020).

2.8. The paper makes a compelling case that the use of ultra-low transmission power and pseudo-random coding enables covert rangefinding. While the low-power operation is evident, the claim of covertness is only speculative. No data or analysis is provided to show how detectable (or undetectable) this system would be in practice against standard detection tools like optical time-domain reflectometers or RF/optical side-channel analyzers.

Response:

We thank the reviewer for raising the concern regarding the claim of covertness.

The covertness we are considering here is the inability of the target to be able to distinguish whether the transmitter is on or off (i.e., are they being ranged). Our system is promising for this, as the intensity of the transmitter light at the target is less than the intensity of the solar background. A tighter constraint on covertness is that the number of photons that are collected at the target in a particular area is less than the fluctuation in photon number from the solar background in the same area. Our system achieves this criterion for the highest solar background levels seen in full daylight with the detailed discussion added in Supplementary Information. For other times of day, we can modify the average number of photons transmitted to be covert, but at night in the absence of background this is difficult to achieve.

As our receiver has a low field of view due to single mode fibre collection, it will only receive scattered light from our transmitter and solar background from the target surface with a comparable illumination area. We see in Figs. 5(h) and 6(b), that the optical signal peaks are fully obscured by the background noise level, demonstrating that the emitted probe remains indistinguishable from ambient background fluctuations on the receiver side.

With respect to the detection tools mentioned, we assume that the reviewer is referring to methods for the target to either detect modulated optical signals from the transmitter or use techniques to discern that the transceiver system is present. Here we have not considered concealing the transceiver, but this could be camouflaged. We consider covertness to be

knowing whether the transceiver is operating or not. In terms of observing the RF modulation of the transmitter, this could be possible with the current system, but true random channel sequences and randomized timing modulation [1], which enable temporal-domain concealment (mentioned on Lines 451-453 of the main manuscript). Spectral concealment can be further enhanced by extending the current 60 nm broadband operation to a wider spectral range through additional channels. From a photon-statistics perspective, the thermal distribution can be mimicked by using a thermal source or coherent pulses with randomly chosen intensities [2].

Finally, we emphasize that covertness is not the main focus of the present work but is discussed in the context of highlighting the potential of our approach for future covert ranging applications. A systematic analysis of covert performance under various operational conditions will be the subject of our forthcoming study. Accordingly, we have revised the relevant discussion and added appropriate references in the revised manuscript.

The revised version:

Lines 441-464: For instance, **in terms of brightness, which is the fundamental prerequisite for all measurements**, the transmission power is set to a low level of 48 μW , with pulse compression applied to obscure the generated correlations. **As shown in Figs. 5(h) and Fig. 6(b), the optical signal peaks are fully concealed within the ambient background noise, with the solar background level nearly three orders of magnitude higher than the signal power, as quantified in Supplementary Section V.** To achieve fully covert operation, the unique signature of the laser source requires further manipulation. For example, the detectable repetition rate can be randomized by introducing random time delays or **being triggered by a true random sequence derived from, for example, shot noise [1], spontaneous Raman scattering [3], chaos laser [4], or a quantum random number generator [5].** Channel indices can also be encoded using a true random sequence. Spectral concealment can be further enhanced by extending the current 60 nm broadband operation to a wider spectral range through additional channels. **Moreover, the laser pulses can be randomly intensity-modulated [2] or replaced with a triggered thermal source to replicate the super-Poissonian statistics of solar background.**

Supplementary Information: add **“Section V. Covertness in Brightness”**:

The covertness we are considering is the inability of the target to be able to distinguish whether the transmitter is on or off (i.e., are they being ranged). Our system is promising for this, as the intensity of the transmitter light at the target is less than the intensity of the solar background. A tighter constraint on covertness is that the number of photons that are collected at the target in a particular area is less than the fluctuation in photon number from the solar background in the same area. In this work, the average source power is 48 μW , producing an illumination spot of approximately 86.7 mm in diameter on the target building (WMB). This corresponds to an irradiance of 0.00813 W/m^2 . By comparison, the solar background irradiance around 1550 nm wavelength is approximately 0.25 $\text{W}/\text{m}^2/\text{nm}$. Assuming the target selects the same 30 nm bandwidth used in our experiment, the effective solar background irradiance is 7.5 W/m^2 , which is three orders of magnitude higher than the

source. While our source power is not comparable to shot-noise fluctuations of the solar background, environmental fluctuations in the solar background are expected to exceed 1% on short time scales and are often found to be over 10% [6]. Thus, our system satisfies this tighter covertness criterion, with the transmitted signal level more than 10 times lower than the background fluctuations.

Ref:

[1] Liu, B., Yu, Y., Chen, Z. & Han, W. True random coded photon counting Lidar. *Opto-Electron. Adv.* **3**, 190044 (2020).

[2] Brougham, T., Samantaray, N. & Jeffers, J. Using random coherent states to mimic quantum illumination. *Phys. Rev. A* **108**, 052404, (2023).

[3] Collins, M. J. *et al.* Random number generation from spontaneous Raman scattering. *Appl. Phys. Lett.* **107**, 141112 (2015).

[4] Hu, Z. *et al.* Chaos single photon LIDAR and the ranging performance analysis based on Monte Carlo simulation. *Opt. Express* **30**, 41658-41670, (2022).

[5] Zhang, Y. *et al.* A simple low-latency real-time certifiable quantum random number generator. *Nat. Commun.* **12**, 1056, (2021).

[6] Kreuwel, F. D. M. *et al.* Analysis of high frequency photovoltaic solar energy fluctuations, *Sol. Energy*, 206, 381-389 (2020).

2.9. The experimental demonstration is undoubtedly impressive, particularly the ability to measure distances with sub-mm accuracy across a 150–400 meter baseline in varying weather and lighting conditions. However, the paper will benefit from additional discussion of environmental robustness — for example, whether alignment drift, thermal effects, or fiber movement degrade performance. Additionally, long-term stability tests and repeatability of precision across independent trials are not discussed.

Response:

We thank the reviewer for the positive assessment of our experimental results and for the valuable comment regarding environmental robustness. We have addressed all related aspects in Supplementary Information Section IV, including the effects of alignment drift, thermal variations, and fiber movement on system performance, as well as long-term stability tests and the repeatability of measurement precision across independent trials.

To summarize, our results demonstrate that the system exhibits strong robustness during day-to-day field operation under various environmental conditions, including nighttime, sunny, cloudy, and rainy periods. This performance is enabled by the noise-reduction capability of the developed source, which remains effective, and even more pronounced, under high

transmission loss and strong background illumination. As a result, performance degradation due to factors such as alignment drift or outdoor fiber movement is negligible. All optical components and connecting fibers are securely mounted and show no noticeable influence on the ranging results.

Continuous 24-hour testing further confirms the system's stability and repeatability, as shown in Fig. S2. Thermal effects are clearly observable; while these can be mitigated through improved thermal insulation, the inherent temperature sensitivity can also serve as a useful feature for tracking ambient temperature variations. Importantly, the measured distance returns to its original value during the following night, demonstrating strong measurement consistency under fluctuating real-world conditions. A minor periodic fluctuation is observed, which is attributed to the FPGA's internal temperature changing and can be reduced with dedicated thermal control.

These findings clearly indicate that the proposed system is inherently robust and well suited for practical field deployment.

The revised version:

Supplementary Information

Section IV. Long-term Stability and Environmental Robustness

Considering long-term stability and repeatability, a systematic evaluation of environmental robustness was performed through continuous 24-hour field testing. As shown in Fig. S2, the measured distance was recorded from midnight to the following midnight with an integration time of 0.1 s and a sampling interval of 6 min. Measurement repeatability was determined from 60 measurements acquired with a 1-second interval, allowing the calculation of the error bars in the plot.

A range fluctuation of approximately 11 mm (corresponding to a ~ 73 ps time drift) with a period of ~ 24 min was observed, which is likely caused by temperature-dependent delay and phase variations in the Zynq-7000 FPGA's internal clocking and routing circuitry. In future work, this will be addressed by using closed-loop temperature control of the FPGA. The overall long-term range tendency is attributed to thermal effects and follows the recorded temperature evolution, as indicated by the average temperature variation trace (red line in Fig. S2 [1]). While such effects can be mitigated using improved thermal isolation on fibre cable, the inherent temperature sensitivity may also serve as a useful feature for tracking environmental temperature changes. Importantly, the measured distance returns to its initial value at night, confirming strong measurement consistency under fluctuating ambient conditions.

Additional factors that may influence system stability are further analysed and discussed below:

1. **Alignment stability:** The optical alignment remained stable throughout day-to-day field measurements and was unaffected by changes in lighting or weather. As shown

in Fig. S2, all optical components are securely fixed on a rigid mounting platform, and the transmitter and receiver share the same multi-axis rotation stage, ensuring structural stability. In addition, the use of Lambertian backscattering relaxes alignment constraints, as the detection is less sensitive to variations in target surface angle.

- Thermal effects:** The fibres in the source and detection modules were housed in a temperature-controlled laboratory environment, where temperature-induced variations are minimal. Only the 300 m outdoor telecom fibre linking the laboratory and the balcony was exposed to environmental temperature changes. As shown in Fig. S2, the range change from 3 a.m. to 3 p.m. is approximately 20 mm, corresponding to a 133.3 ps timing difference and an 11.4 K temperature change with a thermal coefficient of $39 \text{ ps}\cdot\text{km}^{-1}\cdot\text{K}^{-1}$ [2]. The hourly average temperature data (red line) from the Bristol Airport weather station [1] indicates a temperature variation of about 4 K ($^{\circ}\text{C}$) over the same period. The discrepancy from the inferred 11.4 K may arise from local temperature differences between the airport and the city-center university buildings. Nevertheless, as shown in Fig. S2, the observed range variation closely follows the general temperature trend recorded around Bristol Airport. The associated attenuation remains below 0.05 dB [3] for temperatures between -60°C and 85°C , which is negligible for this system, especially as the noise-reduction performance remains effective, and even more pronounced, under high transmission loss conditions.
- Fiber movement:** The system is inherently resilient to fiber-induced disturbances. The noise-suppression performance remains strong and becomes even more pronounced, under high-loss and high-background conditions. Moreover, phase and polarization fluctuations do not impact the field trial, as the system does not rely on interferometric sensitivity. Consequently, bending losses, power variations, and external perturbations introduce no observable degradation during field trials.

Fig. S2 | 24-hour continuous field testing. Detected range (Blue) and hourly average temperature (Red) as a function of time over the 24-hour measurement period, from midnight to the following midnight. The range is measured with an integration time of 0.1 s. Error bars represent the standard deviation of 60 independent range measurements acquired within

each 1-second time window. The range is found at a sampling interval of 6 min. The hourly average temperature data were recorded from the Bristol Airport weather station from WeatherSpark.com [1]. Nighttime and civil twilight are indicated by shaded background overlays.

Ref:

[1] Data downloaded from <https://weatherspark.com/h/d/39587/2024/10/12/Historical-Weather-on-Saturday-October-12-2024-in-Bristol-United-Kingdom>

[2] Slavík R., Marra G., Fokoua E. N., *et al.* Ultralow thermal sensitivity of phase and propagation delay in hollow core optical fibres. *Sci. Rep.* **5**(1), 15447 (2015).

[3]PDF available at

https://lewinb.net/posts/12_what_every_programmer_should_know_about_optical_fiber/P1-1470-AEN.pdf.